# SCALECRAFTER: TUNING-FREE HIGHER-RESOLUTION VISUAL GENERATION WITH DIFFUSION MODELS

**Yingqing He**[*1,3], **Shaoshu Yang**[*2,3], **Haoxin Chen**[3], **Xiaodong Cun**[3], **Menghan Xia**[3],
**Yong Zhang**[†3], **Xintao Wang**[3], **Ran He**[2], **Qifeng Chen**[†1], **Ying Shan**[3]

[1]Hong Kong University of Science and Technology
[2]Chinese Academy of Sciences
[3]Tencent AI Lab

## ABSTRACT

In this work, we investigate the capability of generating images from pre-trained diffusion models at much higher resolutions than the training image sizes. In addition, the generated images should have arbitrary image aspect ratios. When generating images directly at a higher resolution, $1024 \times 1024$, with the pre-trained Stable Diffusion using training images of resolution $512 \times 512$, we observe persistent problems of object repetition and unreasonable object structures. Existing works for higher-resolution generation, such as attention-based and joint-diffusion approaches, cannot well address these issues. As a new perspective, we examine the structural components of the U-Net in diffusion models and identify the crucial cause as the limited perception field of convolutional kernels. Based on this key observation, we propose a simple yet effective *re-dilation* that can dynamically adjust the convolutional perception field during inference. We further propose the dispersed convolution and noise-damped classifier-free guidance, which can enable *ultra-high-resolution* image generation (*e.g.,* $4096 \times 4096$). Notably, our approach *does not require any training or optimization*. Extensive experiments demonstrate that our approach can address the repetition issue well and achieve state-of-the-art performance on higher-resolution image synthesis, especially in texture details. Our work also suggests that a pre-trained diffusion model trained on low-resolution images can be directly used for high-resolution visual generation without further tuning, which may provide insights for future research on ultra-high-resolution image and video synthesis. More results are available at the project website: https://yingqinghe.github.io/scalecrafter/.

## 1 INTRODUCTION

In recent two years, the rapid development of image synthesis has attracted tremendous attention from both academia and industry, especially the most popular text-to-image generation models, such as Stable Diffusion (SD) (Rombach et al., 2022), SD-XL (Podell et al., 2023), Midjourney (Mid), and IF (IF). However, the highest resolution of these models is $1024 \times 1024$, which is far from the demand of applications such as advertisements.

Directly sampling an image with a resolution beyond the training image sizes of those models will encounter severe object repetition issues and unreasonable object structures. As shown in Fig. 1, when using a Stable Diffusion (SD) model trained on images of $512 \times 512$, to sample images of $512 \times 1024$ and $1024 \times 1024$ resolutions, the object repetition appears. The larger the image size, the more severe the repetition.

A few methods attempt to generate images with a larger size than the training image size of SD, *e.g.,* Multi-Diffusion (Bar-Tal et al., 2023) and SyncDiffusion (Lee et al., 2023). In Multi-Diffusion, images generated from multiple windows are fused using the averaged features across the windows

---

[*]Equal Contribution
[†]Corresponding Authors

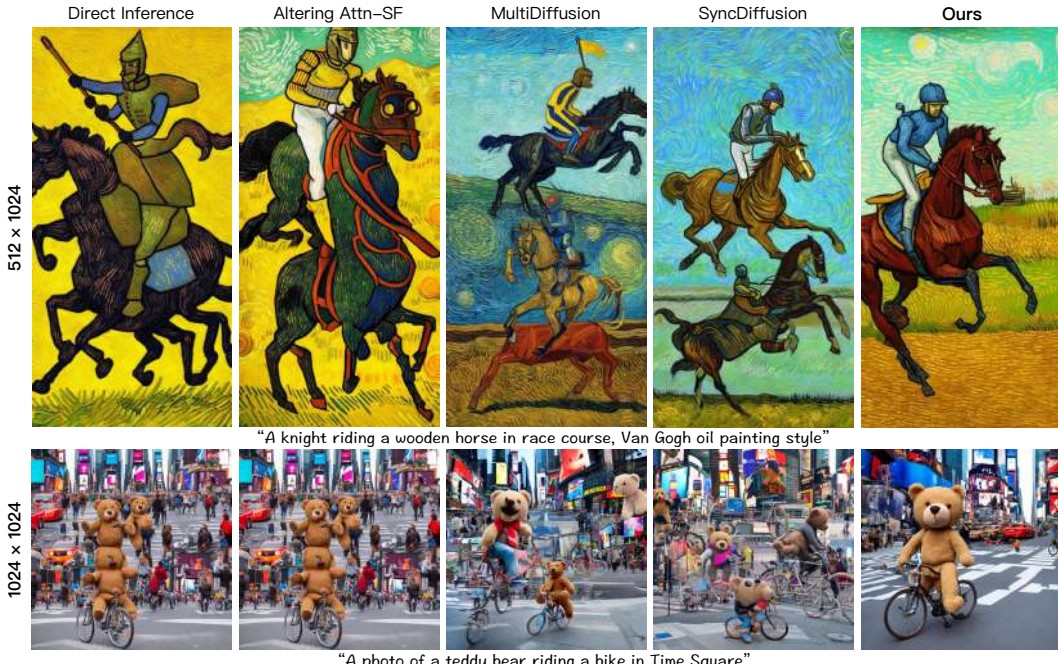

| Direct Inference | Altering Attn–SF | MultiDiffusion | SyncDiffusion | **Ours** |

512 × 1024

"A knight riding a wooden horse in race course, Van Gogh oil painting style"

1024 × 1024

"A photo of a teddy bear riding a bike in Time Square"

Figure 1: Structure repetition issue of higher-resolution generation (Train: $512^2$; Inference: $512 \times 1024$ and $1024^2$). Altering the scaling factor of attention (Jin et al., 2023), and joint diffusion approaches including MultiDiffusion (Bar-Tal et al., 2023) and SyncDiffusion (Lee et al., 2023) fails to address this problem. While our simple *re-dilation* successfully solves this problem and yields structure and semantic correct images, and at meanwhile *require no optimization and tuning cost*.

in all the reverse steps. While SyncDiffusion improves the style consistency of Multi-Diffusion by using an anchor window. However, they focus on the smoothness of the overlap region and cannot solve the repetition issue, as shown in Fig. 1. Most recently, (Jin et al., 2023) studies the SD adaptation for variable-sized image generation through the view of attention entropy. However, their method has a negligible effect on the object repetition issue when increasing the inference resolution.

To investigate the pattern repetition, we sample a set of images of $1024^2$ and $512^2$ from the pre-trained SD model trained with $512^2$ images for comparison. Zooming in on the images, we observe that the images of $1024^2$ have no blur effects and the image quality does not degenerate like the Bilinear upsampling, though their object structures become worse. It indicates that the pre-trained SD model has the potential to generate higher-resolution images without sacrificing image definition.

We then delve into the structural components of SD to analyze their influence, *e.g.,* convolution, self-attention, cross-attention, etc. Surprisingly, when we change convolution to dilated convolution in the whole U-Net using the pre-trained parameters, the overall structure becomes reasonable, *i.e.,* the object repetition disappears. However, the repetition happens to local edges. We then carefully analyze *where*, *when*, and *how* to apply the dilated convolution, *i.e.,* the influence of U-Net blocks, timesteps, and dilation radius. Based on these studies, we propose a tuning-free dynamic re-dilation strategy to solve the repetition. However, as the resolution further increases (e.g., 16x), the decreased generated quality and denoising ability arise. To tackle it, we then propose novel dispersed convolution and noise-damped classifier-free guidance for ultra-high-resolution generation.

Our main contributions are as follows:

- We observe that the primary cause of the object repetition issue is the limited convolutional receptive field rather than the attention token quantity, providing a new viewpoint compared to the prior works.

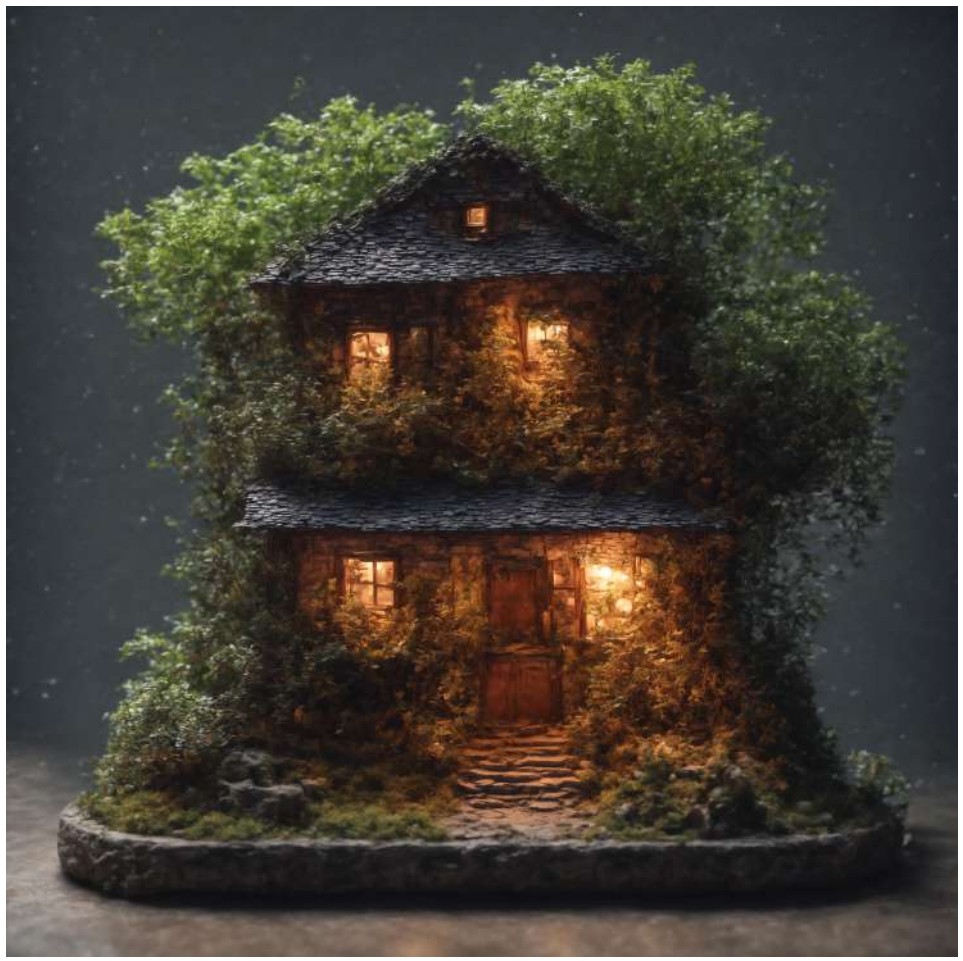

Figure 2: Our method can generate $4096 \times 4096$ images, $16\times$ higher than the training resolution.

- Based on this observation, we propose the simple yet effective *re-dilation* for dynamically increasing the receptive field during inference time. We also propose *dispersed convolution* and *noise-damped classifier-free guidance* for ultra-high-resolution generation.
- We empirically evaluate our approach on various diffusion models, including different versions of Stable Diffusion, and a text-to-video model, with varying image resolutions and aspect ratios, demonstrating the effectiveness of our model.

## 2 RELATED WORK

### 2.1 TEXT-TO-IMAGE SYNTHESIS

Text-to-image synthesis has gained remarkable attention recently due to its impressive generation performance (Saharia et al., 2022; Ramesh et al., 2022; Chang et al., 2023; Ding et al., 2022; Rombach et al., 2022). Among various generative models, diffusion models are popular for their high-quality generation capabilities (Xing et al., 2023; He et al., 2022b; Ho et al., 2022a; He et al., 2023). Following the groundbreaking work of DDPM (Ho et al., 2020), numerous studies have focused on diffusion models for image generation (Nichol et al., 2021; Nichol & Dhariwal, 2021; Kingma & Ba, 2014; Ho et al., 2022b; Gong et al., 2023; Ramesh et al., 2022; Gu et al., 2022).In particular, Latent diffusion models (LDM) (Rombach et al., 2022) have become widely used, as their compact latent space improves model efficiency. Subsequently, a series of Stable Diffusion (SD) models are open-sourced, building upon LDMs and offering high sample quality and creativity (Ma et al., 2023; Chen et al., 2023). Despite their impressive synthesis capabilities, the resolution remains limited to

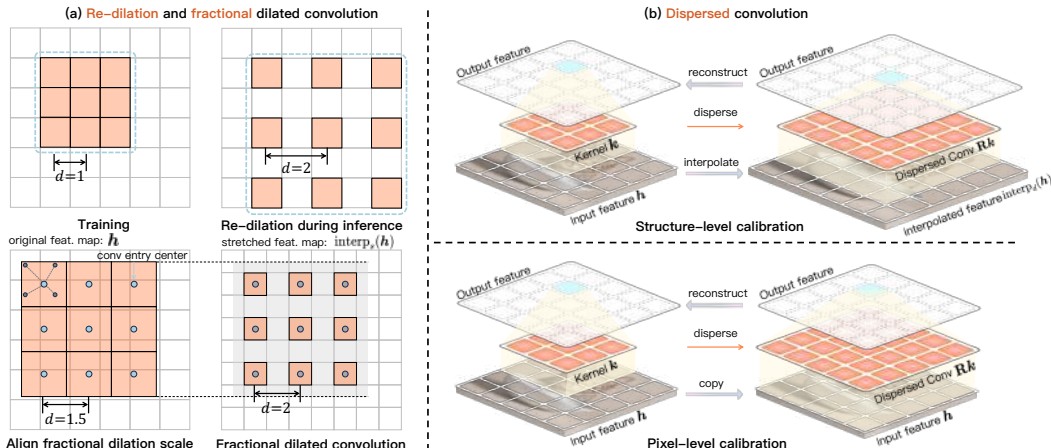

Figure 3: (a) The first row shows re-dilation. Given a pre-trained kernel trained on low-resolution data, we fix the parameters and insert spaces into kernel elements during test time. The second row shows fractional dilated convolution. For each entry of the convolution kernel, we compute the input feature with features near the kernel entry center with bilinear interpolation. This is equivalent to stretch input feature maps and uses a rounded-up dilation scale before the convolution operation. (b) Dispersed convolution can enlarge a pre-trained kernel with a specific scale. We use structure-level calibration to adapt to a new perception field when the input feature dimension is larger and use pixel-level calibration to preserve local information processing ability.

the training resolution, *e.g.*, $512^2$ for SD 2.1 and $1024^2$ for SD XL, necessitating a mechanism for higher resolution generation (*e.g.*, 2K, 4K, etc.).

## 2.2 HIGH-RESOLUTION SYNTHESIS AND ADAPTATION

High-resolution image synthesis is challenging due to the difficulty of learning from higher-dimensional data and the substantial requirement of computational resources. Prior work can mainly be divided into two categories: *training from scratch* (Teng et al., 2023; Hoogeboom et al., 2023; Chen, 2023) and *fine-tuning* (Zheng et al., 2023; Xie et al., 2023). Most recently, (Jin et al., 2023) studies a training-free method for variable-sized adaptation. However, it fails to tackle the higher-resolution generation. Multi-Diffusion (Bar-Tal et al., 2023) and SyncDiffusion (Lee et al., 2023) focus on smoothing the overlap region to avoid inconsistency between windows. However, object repetition still exists in their results. MultiDiffusion can avoid repetition by using the region and text conditions given by users. However, those extra inputs are not available in the scenario of text-to-image generation. Differently, we propose a tuning-free method from the view of the network structure that can fundamentally solve the repetition issue in higher-resolution image synthesis.

## 3 METHOD

### 3.1 PROBLEM FORMULATION AND MOTIVATION

Without the loss of generality, the formulation of this paper considers the $\epsilon$-prediction paradigm of diffusion models. Given a base diffusion model $\boldsymbol{\epsilon}_\theta(\cdot)$ parameterized by $\theta$. It is trained on a fixed pre-defined low-resolution image $\boldsymbol{x} \in \mathbb{R}^{3 \times h \times w}$, our goal is to adapt the model to $\tilde{\boldsymbol{\epsilon}}_\theta(\cdot)$ in a training-free manner to synthesize higher resolution images $\tilde{\boldsymbol{x}} \in \mathbb{R}^{3 \times H \times W}$.

A previous work (Jin et al., 2023) attributes the degradation of performance to the changse in the number of attention tokens and proposes to scale the features in the self-attention layer according to the input resolution. However, when applying it to generate $1024^2$ images, the object repetition is still there (see Fig. 1). We observe that the local structure of each repetitive object seems reasonable and the unreasonable part is the object number when the resolution increases. This encourages us to investigate whether the receptive field of any network component does not fit the larger resolution.

Hence, we modify the components of the SD U-Net to analyze their influence, such as attention, convolution, normalization, etc. We develop *re-dilation* to dissect the effect of the receptive field of attention and convolution, respectively. Re-dilation aims to adjust the network receptive field on a higher-resolution image to maintain the same as the original lower-resolution generation. For re-dilated attention, we partition the feature map into slices with each slice collected via a feature dilation having the same token quantity as training. Then, these slices are fed into the QKV attention in parallel, after which they are merged into the original arrangement. Details are illustrated in the supplementary. However, maintaining the receptive field of attention yields results with indistinguishable differences compared with direct inference. Differently, when increasing the receptive field of convolution in all blocks of the U-Net, fortunately, we observe that the number of objects is correct though there are many artifacts such as noisy background and repetitive edges. Based on the observation, we then develop a more elaborate re-dilation strategy considering where, when, and how to apply dilated convolution.

## 3.2 RE-DILATION

Note that we ignore the feature channel dimension and convolution bias for simplicity in the following. Considering a hidden feature $h \in \mathbb{R}^{m \times n}$ before a convolution layer $f_k(\cdot)$ of the network. Given the convolution kernel $k \in \mathbb{R}^{r \times r}$ and the dilation operation $\Phi_d(\cdot)$ with factor $d$. The dilated convolution $f_k^d(\cdot)$ is computed with

$$f_k^d(h) = h \circledast \Phi_d(k), \ (h \circledast \Phi_d(k)) = \sum_{s+d \cdot t = p} h(p) \cdot k(q), \tag{1}$$

where $p, q$ are spatial locations used to index the feature and kernel, and $\circledast$ denotes convolution operation. Notably, different from traditional dilated convolutions, which share a common dilation factor during training and inference. Our proposed approach dynamically adjusts the dilation factor *only in inference time*, leading us to term it as *re-dilation*. Since the dilation factor can only be an integer, traditional dilated convolution cannot address a fractional multiple of the perception field (i.e., 1.5×). We propose *fractional re-dilated convolution*. Without a loss of information, we round up the target scale to an integer dilation factor and stretch the input feature map to a size where the perception field meets the requirement. Specifically, let $s$ denote the stretch scale and let $\text{interp}_s(\cdot)$ denote a resizing interpolation function (*i.e.*, bilinear interpolation) with scale $s$. We upsample the feature $h$ with the stretch scale to $\text{interp}_s(h)$. The new re-dilation supporting fractional dilation factors is computed as follows:

$$f_k^d(h) = \text{interp}_{1/s} \left( \text{interp}_s(h) \circledast \Phi_{\lceil d \rceil}(k) \right), \ s = \lceil d \rceil / d, \tag{2}$$

where $\lceil \cdot \rceil$ is the round-up operator. A visualization of re-dilated convolution is shown in Fig. 3. Considering the properties of the diffusion model with $T$ timesteps and $L$ layers, we further generalize the re-dilation factor to become layer-and-timestep-aware, yielding $d = D(t, l)$, where $t \in [0, T - 1], l \in [0, L - 1]$, and $D$ is a pre-defined dilation schedule function. Empirically, we find that the re-dilation achieves better synthesis quality when the dilation radius is progressively decreased from deep layers to shallow layers, as well as from noisier steps to less noisy steps, than the fixed dilation factor across all timesteps and layers.

## 3.3 CONVOLUTION DISPERSION

While the aforementioned simple re-dilation effectively handles the generation of higher-resolution generation at 4× resolution, it suffers from the periodic downsampling problem (e.g., grinding artifacts) (Wang & Ji, 2018), *i.e.*, the features will not consider information from different dilated convolution splits, which leads to the generated image appears repetitive edges under a much higher resolution. This problem arises when adapting a diffusion model to generate a much higher resolution, e.g., 8× and 16×. To alleviate the problem, we propose to increase the receptive field of a pre-trained convolution layer by dispersing its convolution kernel. Our *convolution dispersion* method is shown in Fig. 3. Given a convolution layer with kernel $k \in \mathbb{R}^{r \times r}$ and a target kernel size $r'$ (if the required perception field multiple is $d$ and r is odd, then $r' = d(r - 1) + 1$), our method applies a linear transform $\mathbf{R} \in \mathbb{R}^{r'^2 \times r^2}$ to get a dispersed kernel $k' = \mathbf{R}k$. We apply *structure-level* and *pixel-level calibration* to enlarge the convolution kernel while keeping the capability of the original convolution layer.

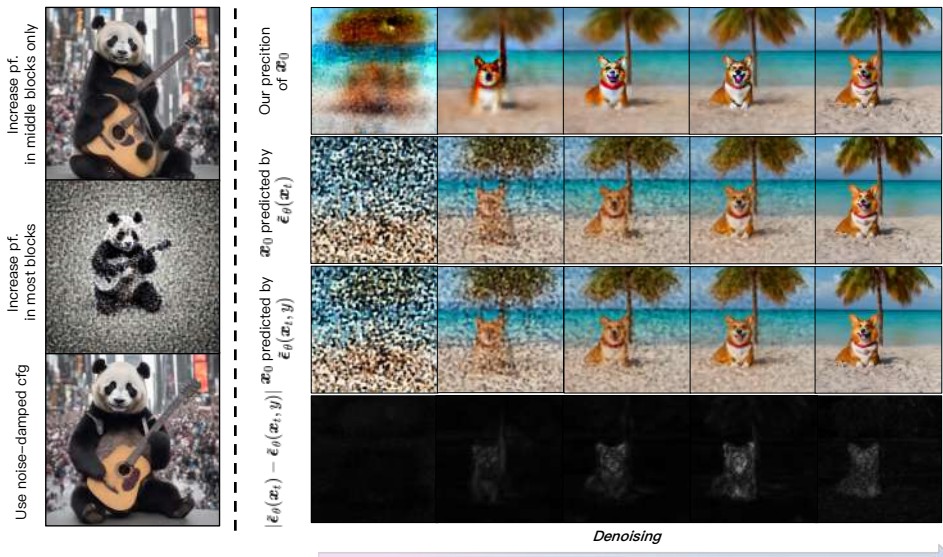

Figure 4: **left**: Samples by increasing perception field in middle blocks and most blocks (middle and outskirt blocks). The middle blocks-only setting fails to produce the correct small object structures. **right**: The first row shows the predicted original sample using noise-damped classifier-free guidance. The second and third rows show the prediction using $\tilde{\epsilon}_\theta(\boldsymbol{x}_t, y)$ and $\tilde{\epsilon}_\theta(\boldsymbol{x}_t)$. $\tilde{\epsilon}_\theta(\boldsymbol{x}_t, y)$ and $\tilde{\epsilon}_\theta(\boldsymbol{x}_t)$ fails to remove noise during sampling. However, their predictions exhibit a very similar noise pattern. The fourth row illustrates $|\tilde{\epsilon}_\theta(\boldsymbol{x}_t, y) - \tilde{\epsilon}_\theta(\boldsymbol{x}_t)|$. The erroneous noise prediction vanishes and we can utilize the remaining useful information.

We use *structure-level calibration* to preserve the performance of a pre-trained convolution layer when the size of the input feature map changes. Consider an arbitrary convolution layer $f_{\boldsymbol{k}}(\cdot)$ and the input feature map $\boldsymbol{h}$. Structure-level calibration requires the following equation:

$$\text{interp}_d(f_{\boldsymbol{k}}(\boldsymbol{h})) = f_{\boldsymbol{k}'}(\text{interp}_d(\boldsymbol{h})), \ \boldsymbol{k}' = \mathbf{R}\boldsymbol{k} \tag{3}$$

where $f_{\boldsymbol{k}'}(\cdot)$ receives the interpolated feature map and keeps its output the same as the interpolated original output $\text{interp}_d(f_{\boldsymbol{k}}(\boldsymbol{h}))$. Eqn. 3 is underdetermined since the enlarged kernel $\boldsymbol{k}'$ has more elements than $\boldsymbol{k}$. To solve this equation, we introduce *pixel-level calibration* to ensure the enlarged new convolution kernel behaves similarly on the original feature map $\boldsymbol{h}$. Mathematically, pixel-level calibration requires $f_{\boldsymbol{k}}(\boldsymbol{h}) = f_{\boldsymbol{k}'}(\boldsymbol{h})$. Then, we combine this with Eqn. 3 to formulate a linear least square problem:

$$\mathbf{R} = \underset{\mathbf{R}}{\arg\min} \|\text{interp}_d(f_{\boldsymbol{k}}(\boldsymbol{h})) - f_{\boldsymbol{k}'}(\text{interp}_d(\boldsymbol{h}))\|_2^2 + \eta \cdot \|f_{\boldsymbol{k}}(\boldsymbol{h}) - f_{\boldsymbol{k}'}(\boldsymbol{h})\|_2^2 \tag{4}$$

where $\eta$ is a weight controlling the focus of dispersed convolution. We derive an enlarged kernel with convolution dispersion by solving the least square problem. Note that $\mathbf{R}$ is not relevant to the exact numerical value of the input feature or the kernel. One can apply it to any convolution kernels to enlarge it from $r \times r$ to $r' \times r'$. Convolution dispersion can be used along with the re-dilation technique to achieve a much larger perceptual field without suffering from the periodic sub-sampling problem. To achieve a fractional perception field scale factor, we replace the dilation operation with convolution dispersion in fractional dilated convolution introduced above.

## 3.4 NOISE-DAMPED CLASSIFIER-FREE GUIDANCE

To sample at a much higher resolution (*i.e.*, $4\times$ in both height and width), we need to increase the perception field in the outer blocks in the denoising U-Net to generate the correct structure in small objects as shown in Fig. 4. However, we find that the outside block in the U-Net contributes a lot to estimating the noise contained in the input. When we try to increase the convolution perceptual field in these blocks, the denoising capability of the model is damaged. As a result, it is challenging to generate the correct small structures while maintaining the denoising ability of the original model.

| Res | Method | SD 1.5 | | | | SD 2.1 | | | | SD XL 1.0 | | | |
|---|---|---|---|---|---|---|---|---|---|---|---|---|---|
| | | $FID_r$ | $KID_r$ | $FID_b$ | $KID_b$ | $FID_r$ | $KID_r$ | $FID_b$ | $KID_b$ | $FID_r$ | $KID_r$ | $FID_b$ | $KID_b$ |
| 4× 1:1 | Direct-Inf | 38.50 | 0.014 | 29.30 | 0.008 | 29.89 | 0.010 | 24.21 | 0.007 | 67.71 | 0.029 | 45.55 | 0.014 |
| | Attn-SF | 38.59 | 0.013 | 29.30 | 0.008 | 28.95 | 0.010 | 22.75 | 0.007 | 68.93 | 0.028 | 46.07 | 0.013 |
| | Ours | **32.67** | **0.012** | **24.93** | **0.007** | **20.88** | **0.008** | **16.67** | **0.005** | **64.75** | **0.024** | **28.15** | **0.009** |
| 6.25× 1:1 | Direct-Inf | 55.47 | 0.020 | 48.54 | 0.015 | 52.58 | 0.018 | 48.13 | 0.014 | 93.91 | 0.041 | 54.90 | 0.020 |
| | Attn-SF | 55.96 | 0.020 | 49.03 | 0.015 | 50.62 | 0.017 | 45.57 | 0.014 | 93.92 | 0.042 | 54.89 | 0.019 |
| | Ours | **52.11** | **0.019** | **45.86** | **0.014** | **33.36** | **0.010** | **30.66** | **0.008** | **80.72** | **0.032** | **47.15** | **0.015** |
| 8× 1:2 | Direct-Inf | 74.52 | 0.032 | 68.98 | 0.027 | 69.89 | 0.029 | 55.48 | 0.020 | 122.41 | 0.062 | 82.51 | 0.037 |
| | Attn-SF | 74.42 | 0.032 | 68.81 | 0.027 | 68.97 | 0.029 | 53.97 | 0.020 | 122.21 | 0.062 | 82.35 | 0.037 |
| | Ours | **58.21** | **0.022** | **52.76** | **0.017** | **58.57** | **0.021** | **49.41** | **0.015** | **119.58** | **0.057** | **50.70** | **0.019** |
| 16× 1:1 | Direct-Inf | 111.34 | 0.046 | 106.70 | 0.042 | 104.70 | 0.043 | 104.10 | 0.040 | 153.33 | 0.070 | 144.99 | 0.061 |
| | Attn-SF | 110.10 | 0.046 | 105.42 | 0.042 | 104.34 | 0.043 | 103.61 | 0.041 | 153.68 | 0.070 | 144.84 | 0.061 |
| | Ours | **78.22** | **0.027** | **65.86** | **0.023** | **59.40** | **0.021** | **57.26** | **0.018** | **131.03** | **0.063** | **124.01** | **0.055** |

Table 1: Quantitative comparisons among different methods for higher-resolution generation.

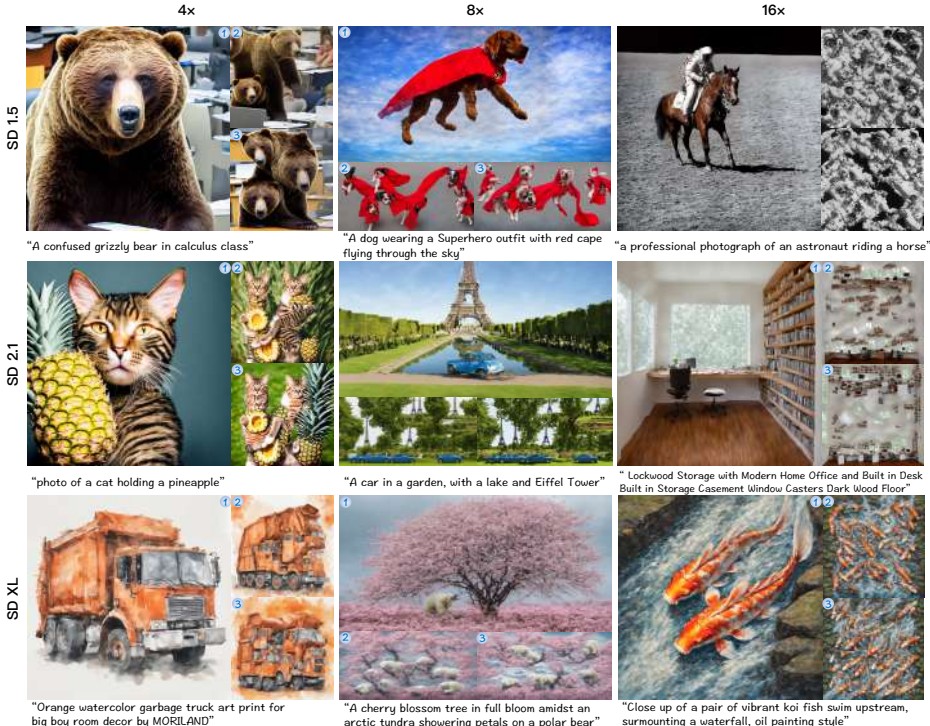

Figure 5: Qualitative comparisons between our method (①), direct inference (②), and altering the attention scaling factor (③) under 4x, 8x, and 16x settings and three different pretrained models.

We propose *noise-damped classifier-free guidance* to address the difficulties. Our method incorporates the two model priors, a model with strong denoising capabilities $\epsilon_\theta(\cdot)$, and a model with re-dilated or dispersed convolution in most blocks that generates correct content structures $\tilde{\epsilon}_\theta(\cdot)$. Then, the sampling is performed via a linear combination of the estimations with a scale $w$:

$$\epsilon_\theta(\boldsymbol{x}_t) + w \cdot (\tilde{\boldsymbol{\epsilon}}_\theta(\boldsymbol{x}_t, y) - \tilde{\boldsymbol{\epsilon}}_\theta(\boldsymbol{x}_t)), \tag{5}$$

where $y$ is the input text prompt. Eqn. 5 includes a base prediction $\epsilon_\theta(\boldsymbol{x}_t)$ that ensures effective denoising during the sampling process. The guidance term $\tilde{\boldsymbol{\epsilon}}_\theta(\boldsymbol{x}_t, y) - \tilde{\boldsymbol{\epsilon}}_\theta(\boldsymbol{x}_t)$ includes two poor noise predictions. However, in Fig. 4, our experiments demonstrate that the erroneous noise predictions in $\tilde{\epsilon}_\theta(\mathbf{x}_t)$ and $\tilde{\epsilon}_\theta(\mathbf{x}_t, y)$ are very similar. Such erroneous noise prediction vanishes in the results of $\tilde{\epsilon}_\theta(\boldsymbol{x}_t, y) - \tilde{\epsilon}_\theta(\boldsymbol{x}_t)$, and the remaining information is useful for generating correct object structures.

## 4 EXPERIMENTS

**Experiment setup.** We conducted evaluation experiments on text-to-image models, Stable Diffusion (SD), including three prevalent versions: SD 1.5 (Rombach et al., 2022), SD 2.1 (Diffusion,

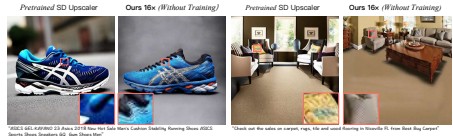

| Figure 6: Visual comparisons with SD-SR. | Figure 7: Qualitative ablation results. |
|---|---|

| Method | $FID_r$-4× | $KID_r$-4× | TD-16× |
|---|---|---|---|
| SD+SR | **12.59** | **0.005** | 38% |
| Ours | 20.88 | 0.008 | **62%** |

| Method | $FVD_r$-4× | $KVD_r$-4× |
|---|---|---|
| Direct Inference | 674.14 | 78.31 |
| **Ours** | **418.80** | **31.78** |

Table 2: Quantitative comparisons with SR. Table 3: Quantitative results on T2V.

2022), and SD XL 1.0 (Podell et al., 2023) in inferring four unseen higher resolutions. We experimented with four resolution settings, which are 4 times, 6.25 times, 8 times, and 16 times more pixels than the training. Specifically, for both SD 1.5 and SD 2.1, the training size is $512^2$, and the inference resolutions are $1024^2$, $1280^2$, $2048\times1024$, $2048^2$, respectively. For SD XL, the training resolution is $1024^2$, and the inference resolutions are $2048^2$, $2560^2$, $4096\times2048$, $4096^2$. We also evaluate our approach on a text-to-video model for 2 times higher resolution generation. Please see our supplementary for the detailed hyperparameter settings.

**Testing dataset and evaluation.** We evaluate performance on the dataset of Laion-5B (Schuhmann et al., 2022) which contains 5 billion image-caption pairs. When the inference resolution is $1024^2$, we sample 30k images with randomly sampled text prompts from the dataset. Due to massive computation, we sample 10k images when the inference resolution is higher than $1024^2$. In main evaluations, we experiment with a normal higher resolution setting (4×, 1:1), a fractional scaling resolution (6.25, 1:1), a varied aspect ratio (8×, 1:2), and an extreme higher resolution (16×, 1:1). In other experiments, we evaluate the metrics with the aspect ratio of 1:1 unless otherwise specified. Following the standard evaluation protocol, we measure the Frechet Inception Distance (FID) and Kernel Inception Distance (KID) between generated images and real images to evaluate the generated image quality and diversity, referred to as $FID_r$ and $KID_r$. We adopt the implementation of clean-fid (Parmar et al., 2022) to avoid discrepancies in the image pre-processing steps. Since the pre-trained models have the capability of compositing different concepts that do not appear in the training set, we also measure the metrics between the generated samples under the base training resolution and inference resolution, referred to as $FID_b$ and $KID_b$. This evaluates how well our method can preserve the model's original ability when sampling under a new resolution.

## 4.1 EVALUATION

**Comparision with training-free methods.** We compare our method with the vanilla text-to-image diffusion model (Direct-Inf) and a tuning-free method (Jin et al., 2023) via altering the attention scaling factor (Attn-SF). As Multi-Diffusion (Bar-Tal et al., 2023) and SyncDiffusion (Lee et al., 2023) cannot alleviate the repetition issue, they are not compared here. Our results are shown in Tab. 1. Compared to baselines, we achieve better scores in all settings, indicating our method preserves the generation ability of a pre-trained diffusion model much better in a new scale. Visual comparisons are shown in Fig. 5. Direct inference and Attn-SF tend to generate repeated contents, resulting in an unnatural image structure. On the contrary, our method can generate plausible structures and highly detailed textures in unseen image resolutions.

**Comparison with the diffusion super-resolution model (SR).** Although our approach does not require any extra datasets or extensive training efforts, to comprehensively evaluate our performance, we compare our approach with a pre-trained Stable Diffusion super-resolution (SD-SR) model: SD 2.1-upscaler-4× (Ups) at 4x and 16x higher resolution. Both our approach and the SR are combined with the SD 2.1-$512^2$. Qualitative and quantitative results are shown in Fig. 6 and Tab. 2. As seen in Fig. 6, our method synthesizes better details and textures, e.g., the shoes and cushion. Note that the calculation of FID and KID requires image downsampling to $229^2$ which cannot measure the

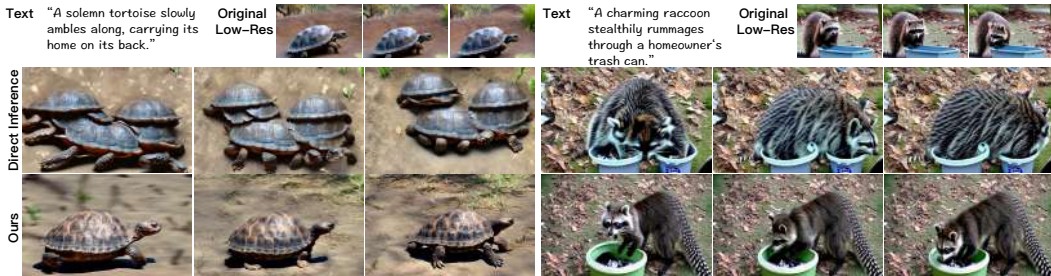

Figure 8: Quantitative results on 4x resolution of T2V.

definition of texture. Hence, we perform a user preference study on the $16 \times$ setting to ask users to choose an image with a better texture definition between ours and SR. The responses to 300 questions are collected. We summarize the results with the percentage of the user's choices, referred to as Texture Definition (TD) (4$^{\text{th}}$ column in the Tab. 2). Though our FID and KID are slightly worse than the pretrained SD-SR, our method synthesizes high-resolution images with a lower-resolution generative model *without any extra training*, while the SR requires substantial data and computation to train and exhibits *worse texture details*. This demonstrates the pretrained SD already learns the rich texture priors. With proper utilization, we can leverage this prior to synthesizing high-quality and higher-resolution images directly.

### 4.2 ABLATION STUDY

| Method | FID$_r$-16× | KID$_r$-16× |
|---|---|---|
| Ours | **78.22** | **0.027** |
| w/o progression | 94.90 | 0.040 |
| w/o conv dispersion | 106.18 | 0.051 |
| w/o nd-cfg | 112.15 | 0.055 |

Table 4: Ablation study on SD 1.5.

We conduct ablation studies on SD 1.5 and $16 \times$ (1:1) resolution to generate $2048^2$. Visual results of removing three key technical components, including timestep-wise progressive re-dilation (progression), convolution dispersion, and noise-damped classifier-free guidance (nd-cfg), can be seen in Fig. 7. It highlights that the progression improves visual quality, especially in image details. Nd-cfg ensures less noise in the final image, and dispersed convolution improves the fidelity of object structures. Tab. 4 shows quantitative results, which show that without these components, the FID dropped by 16.68, 27.96, and 33.93, respectively. Hence, every technical component brings considerable improvements.

### 4.3 APPLY ON VIDEO DIFFUSION MODELS

To verify the generalization ability of our method for video generation models, we apply it to a pre-trained text-to-video model, LVDM (He et al., 2022a). As shown in Fig. 8, our method can generate higher-resolution videos without image definition degeneration. The quantitative results are in Tab. 3. Metrics are computed using the video counterpart Frechet Video Distance (FVD) (Unterthiner et al., 2018) and Kernel Video Distance (KVD) (Unterthiner et al., 2019) with 2048 sampled videos and are evaluated on the Webvid-10M (Bain et al., 2021).

### 5 CONCLUSIONS

We investigate the possibility of sampling images at a much higher resolution than the training resolution of pre-trained diffusion models. Directly sampling a higher-resolution image can preserve the image definition but will encounter severe object repetition issues. We delve into the architecture of the SD U-Net and explore the receptive field of its components. Fortunately, we observe that convolution is critical for sampling higher-resolution images. We then propose an elaborate dynamic re-dilation strategy to remove the repetition and also propose the dispersed convolution and noise-damped classifier-free guidance for ultra-high-resolution generation. Evaluations are conducted to demonstrate the effectiveness of our methods for different text-to-image and text-to-video models.

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

## 6 APPENDIX

## A EXPERIMENT DETAILS

### A.1 MODEL LAYERS IN OUR METHOD

The U-Net of Stable Diffusion (SD) v1.5, SD 2.1, and SD XL 1.0 share a similar convolution layer layout. We explain which layer to use re-dilated or dispersed convolutions without a loss of generality. We follow the naming of layers in diffuers[1]. A list of convolution layers contained in a U-Net block is shown in Tab. 5. The attention projection layers and convolution shortcut layers will not use re-dilation or dispersion since the convolution kernel in these layers is $1 \times 1$. Note that the first and the last convolution in the U-Net (conv_in and conv_out) will not use our method since they do not contribute to generating image contents. Also, the spatial part of the text-to-video model we used shares the same architecture as the SD. Therefore, layers of the following mentioned are also the same as our video experiment.

| Layer name | Exist in all blocks | Use our method |
|---|---|---|
| attentions.0.proj_in | ✓ | ✗ |
| attentions.0.proj_out | ✓ | ✗ |
| attentions.1.proj_in | ✗ | ✗ |
| attentions.1.proj_out | ✗ | ✗ |
| attentions.2.proj_in | ✗ | ✗ |
| attentions.2.proj_out | ✗ | ✗ |
| resnets.0.conv1 | ✓ | ✓ |
| resnets.0.conv2 | ✓ | ✓ |
| resnets.0.conv_shortcut | ✗ | ✗ |
| resnets.1.conv1 | ✓ | ✓ |
| resnets.1.conv2 | ✓ | ✓ |
| downsamplers.0.conv | ✗ | ✓ |

Table 5: The layers to use our method in a U-Net block. The second column shows the existence condition since some layers cannot be seen in specific U-Net blocks.

### A.2 HYPERPARAMETERS

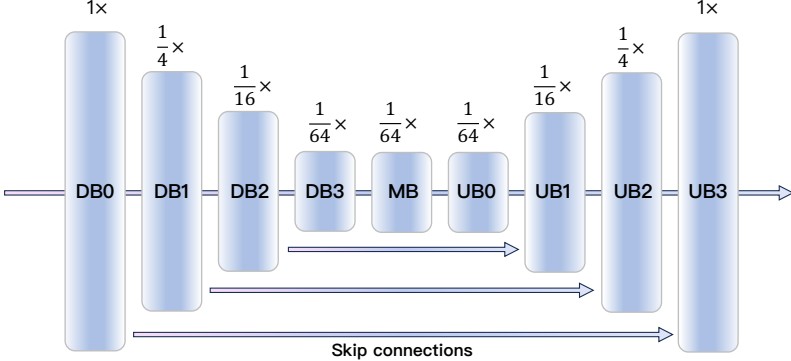

Figure 9: Reference block names in the following experiment details. The fractional multiples above blocks are the spatial pixel number of feature maps within the block compared to the network input. i.e, the input latent has $64^2$ spatial dimension, then the size of feature maps in DB3 to UB0 is $8^2$.

---

[1]https://github.com/huggingface/diffusers

We explain our selection for hyperparameters in this section. All samples are generated using the default classifier-free guidance scale of the corresponding pre-trained model (i.e. SD 1.5 and SD 2.1 use 7.5, SD XL 1.0 uses 5.0). Our SD 2.1 experiments use a similar setting to SD 1.5. We list the hyperparameters for SD 1.5 only for brevity. The evaluation settings for SD 1.5 are shown in Tab. 6, 7, 8, 9. The settings for SD XL 1.0 are shown in Fig. 10, 11, 12, 13. A reference for block names and their exact location in the U-Net can be found in Fig. 9. The tables show detailed settings about which block to use re-dilation conv and dispersed conv. Dilation scale rb. means the dilation scale for re-dilated blocks and dilation scale db. defines the dilation scale for dispersed blocks. If the sampling uses noise-damped classifier-free guidance, we construct a $\epsilon_\theta(\cdot)$ with strong denoising capability by turning some outskirt blocks that use re-dilated and dispersed convolution to the original blocks. The chosen ones that become the original blocks are listed in noise-damped blocks.

| Params | Values |
|---|---|
| latent resolution | $4\times128\times128$ |
| re-dilated blocks | [DB3, MB, UB0] |
| dilation scale rb. | $[2, 2, 2]$ |
| dispersed blocks | $\emptyset$ |
| progressive | ✗ |
| noise-damped cfg. | ✗ |
| inference timesteps | 50 |
| $\tau$ | 30 |

Table 6: $1024^2$ SD 1.5 experiment settings.

| Params | Values |
|---|---|
| latent resolution | $4\times160\times160$ |
| re-dilated blocks | [DB3, MB, UB0] |
| dilation scale rb. | $[2.5, 2.5, 2.5]$ |
| dispersed blocks | $\emptyset$ |
| progressive | ✗ |
| noise-damped cfg. | ✗ |
| inference timesteps | 50 |
| $\tau$ | 30 |

Table 7: $1280^2$ SD 1.5 experiment settings.

| Params | Values |
|---|---|
| latent resolution | $4\times128\times256$ |
| re-dilated blocks | [DB0, DB1, DB2, DB3, MB, UB0, UB1, UB2, UB3] |
| dilation scale rb. | $[2, 2, 2, 2, 2, 2, 2, 2, 2]$ |
| dispersed blocks | $\emptyset$ |
| progressive | ✗ |
| noise-damped cfg. | ✓ |
| noise-damped blocks | [DB0, DB1, DB2, UB1, UB2, UB3] |
| inference timesteps | 50 |
| $\tau$ | 30 |

Table 8: $2048\times1024$ SD 1.5 experiment settings.

| Params | Values |
|---|---|
| latent resolution | $4\times256\times256$ |
| re-dilated blocks | [DB0, DB1, UB2, UB3] |
| dilation scale rb. | $[2, 4, 4, 2]$ |
| dispersed blocks | [DB2, DB3, MB, UB0, UB1] |
| dilation scale db. | $[2, 2, 2, 2, 2]$ |
| dispersed kernel size | $3 \times 3 \to 5 \times 5$ |
| progressive | ✓ |
| noise-damped cfg. | ✓ |
| noise-damped blocks | [DB0, DB1, UB2, UB3] |
| inference timesteps | 50 |
| $\tau$ | 35 |

Table 9: $2048^2$ SD 1.5 experiment settings.

## A.3 SYNCHRONIZE STATISTICS BETWEEN TILES IN GROUPNORM

| Params | Values |
|---|---|
| latent resolution | 4×256×256 |
| re-dilated blocks | [DB3, MB, UB0] |
| dilation scale rb. | [2, 2, 2] |
| dispersed blocks | ∅ |
| progressive | ✗ |
| noise-damped cfg. | ✗ |
| inference timesteps | 50 |
| $\tau$ | 30 |

Table 10: $2048^2$ SD XL 1.0 settings.

| Params | Values |
|---|---|
| latent resolution | 4×320×320 |
| re-dilated blocks | [DB1, DB2, DB3, MB, UB0, UB1, UB2] |
| dilation scale rb. | [2, 2, 2.5, 2.5, 2.5, 2, 2] |
| dispersed blocks | ∅ |
| progressive | ✗ |
| noise-damped cfg. | ✓ |
| noise-damped blocks | [DB1, DB2, UB1, UB2] |
| inference timesteps | 50 |
| $\tau$ | 30 |

Table 11: $2560^2$ SD XL 1.0 experiment settings.

| Params | Values |
|---|---|
| latent resolution | 4×256×512 |
| re-dilated blocks | [DB1, DB2, DB3, MB, UB0, UB1, UB2] |
| dilation scale rb. | [2, 2, 2, 2, 2, 2, 2] |
| dispersed blocks | ∅ |
| progressive | ✗ |
| noise-damped cfg. | ✓ |
| noise-damped blocks | [DB1, DB2, UB1, UB2] |
| inference timesteps | 50 |
| $\tau$ | 30 |

Table 12: 4096×2048 SD XL 1.0 experiment settings.

| Params | Values |
|---|---|
| latent resolution | 4×512×512 |
| re-dilated blocks | [DB2, UB1] |
| dilation scale rb. | [2, 2] |
| dispersed blocks | [DB3, MB, UB0] |
| dilation scale db. | [2, 2] |
| dispersed kernel size | $3 \times 3 \to 5 \times 5$ |
| progressive | ✓ |
| noise-damped cfg. | ✓ |
| noise-damped blocks | [DB2, UB1] |
| inference timesteps | 50 |
| $\tau$ | 35 |

Table 13: $4096^2$ SD XL settings.

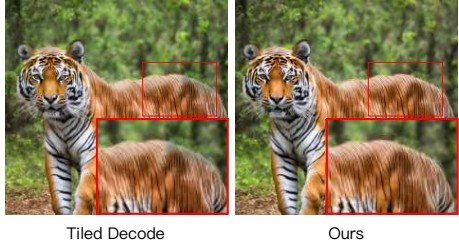

Figure 10: Direct tiled decode causes abrupt changes in tile borders and different color tones in tiles. We synchronize the statistics in VAE GroupNorm between tiles to address this problem.

When the generated image size is large (i.e., $> 2048 \times 2048$), the VAE of SD requires enormous VRAM for decoding and is usually not applicable on a personal GPU. A simple solution is decoding in tiles. However, tiled decoding usually causes abrupt changes between different tiles as shown in Fig. 10. To solve this, one can make overlapped regions between tiles and interpolate on the overlapped regions. However, another problem of tiled decoding is the inconsistent color tone between tiles. We figure out this is caused by the independent computation of GroupNorm (GN) layers in VAE between tiles. We propose to synchronize the feature statistics in GN in different tiles. Specifically, we compute the mean and std using all tiles instead of using only current ones. As shown in Fig. 10, it eliminates the color tone difference efficiently.

## B  RE-DILATED ATTENTION

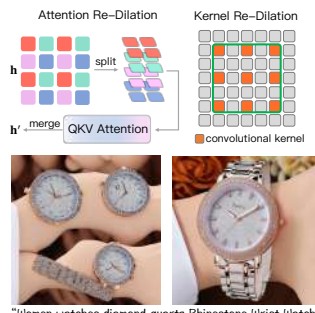

Figure 11: Illustration and results of two re-dilations.

Here, we introduce the experimented re-dilated attention. We aim to keep the original receptive field of attention, e.g., the attention token quantity. Thus, before calculating the attentional features, we first split the input feature map into four slices (the resolution is 4x higher than the training), and for each slice, we flat them into token sequences and feed them into the QKV attention. After the attention calculation, we merge them back to form the original feature arrangement. This operation strictly controls the token length of attention to be the same as training. However, this cannot solve the structure issue of the generated image, as shown in the 2nd row of Fig. 11. However, when applying the redilation on the convolutional kernel, the structure is totally correct. This demonstrates that the key cause of structure repetition lies in convolutional kernels.

## C   MORE COMPARISONS

| Method | FID$_r$ | KID$_r$ | pFID$_r$ | pKID$_r$ | sFID$_r$ | sKID$_r$ | Time (s) | #param |
|---|---|---|---|---|---|---|---|---|
| SD XL | 18.50 | 0.005 | 29.63 | 0.014 | 16.68 | 0.007 | 6.5 | 3.5B |
| SD 2.1+SR | 15.39 | 0.005 | **17.30** | **0.005** | 14.57 | 0.007 | 9.5 (1.5+8) | 2.2B |
| Ours (SD 2.1) | 18.73 | 0.005 | 20.97 | **0.005** | **10.17** | **0.004** | **5.6** | **1.3B** |
| Ours (SD 2.1)+ LR | **9.96** | **0.003** | 19.27 | 0.007 | 11.05 | **0.004** | 6.3(1.5+4.8) | **1.3B** |

Table 14: Comparison results with state-of-the-art image generation models and super-resolution (SR) model under the resolution of $1024^2$. Time indicates the second used for synthesizing one image on one A100 GPU with 16-bit precision). #param stands for the number of model parameters. pFID$_r$ and pKID$_r$ represent patch-FID and patch-KID, respectively. pFID$_r$ Chai et al. (2022), pKID$_r$ Chai et al. (2022), sFID$_r$ Nash et al. (2021), and sKID$_r$ Nash et al. (2021) are used to measure the texture details of generated samples.

We make a comprehensive comparison regarding general generation quality (FID$_r$, KID$_r$), texture details (pFID$_r$, pKID$_r$, sFID$_r$, sKID$_r$), inference time and number of model parameters with SD+SR and high-resolution image generation method SD XL, as shown in Tab. 14. Specifically, pFID/pKID avoids the downsampling operation and instead uses cropping in the metric calculation. sFID/sKID uses the features before the global average pooling to retain low-level details in the feature for the metric calculation, as well as avoids downsampling. The evaluation dataset is a 30k subset from Laion-5B with a resolution larger than $1024^2$.

Results show that our training-free method (with no low-resolution reference image) achieves almost the comparable generation performance compared with well-trained SD+SR. Additionally, we achieve better texture details than SD + SR (see the sFID and sKID metrics, as well as the user study in the main paper). With the low-resolution generated samples as guidance, our method achieves much better results than SD+SR. At the same time, our method has 59% inference time and 59% model parameters less than SD+SR, showing our better efficiency. Compared with SD XL, we achieve both better metrics and lower inference time and parameter numbers.

## D   MORE ABLATIONS

We conduct additional ablation experiments to investigate the impact of increasing inference resolutions. As depicted in Figure 12, it is evident that as the inference resolution increases, the degree of structure distortion becomes more pronounced. Despite these challenges, our method is capable of effectively addressing and mitigating these issues, even in extremely demanding settings.

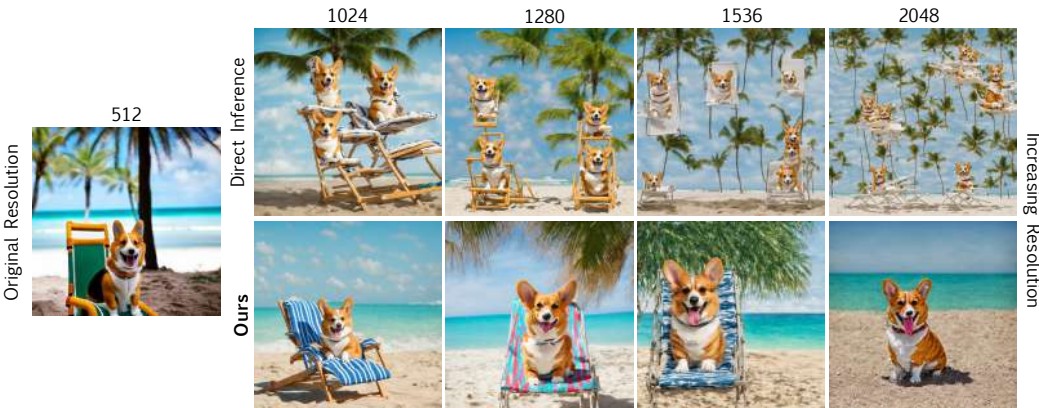

Figure 12: Performance change when increasing the image resolution. When increasing the resolution, the problem becomes more challenging. Our method is still capable of addressing these issues and maintaining a correct image structure.

## E    LIMITATIONS

Ensuring the accuracy of the local structure remains challenging, as demonstrated in Figure 13, particularly with regard to intricate details like the fingers of the robot and the legs of the chairs. It is worth noting that this issue is not exclusive to our method but also exists in the original lower-resolution model. As our approach is training-free, it inherits the limitations of the original model.

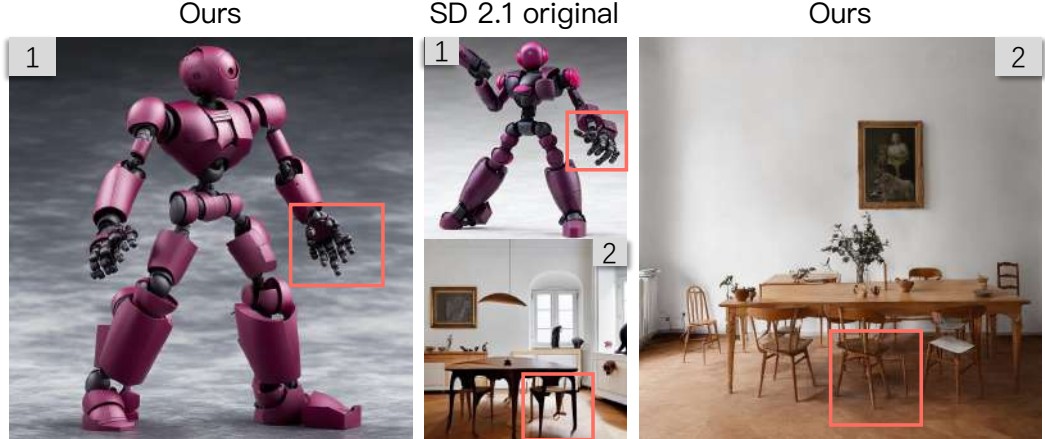

Figure 13: Failure cases on local structures. Our method has failures in the local structure of the generated image. We observed that the original lower-resolution model also struggled with this problem.

## F    CLOSED-FROM SOLUTION FOR DISPERSED CONVOLUTION

We follow the notation in our main paper. Given a convolution layer with kernel $\boldsymbol{k} \in \mathbb{R}^{r \times r}$ and a target kernel size $r'$. We find a dispersion transform $\mathbf{R} \in \mathbb{R}^{r'^2 \times r^2}$ to get a dispersed kernel $\boldsymbol{k}'$. Considering an input feature map $\boldsymbol{h} \in \mathbb{R}^{n \times n}$, we ignore the bias in convolution, since it will not influence our results. Then structure-level calibration and pixel-level calibration can form an equation set:

$$\text{interp}_d(f_k(\boldsymbol{h})) = f_{\boldsymbol{k}'}(\text{interp}_d(\boldsymbol{h}))$$
$$\eta f_{\boldsymbol{k}}(\boldsymbol{h}) = \eta f_{\boldsymbol{k}'}(\boldsymbol{h}), \tag{6}$$

where $\text{interp}_d(f_k(\boldsymbol{h})) \in \mathbb{R}^{nd \times nd}, f_{\boldsymbol{k}}(\boldsymbol{h}) \in \mathbb{R}^{n \times n}$. The equation set has $(nd)^2 + n^2$ equations. Each equation is made up of the sum of terms $k_{ij}h_{ij}$ (elements in $\boldsymbol{k}$ and $\boldsymbol{h}$) where the coefficient of the terms are linear combinations of $R_{ij}$ (elements in $\mathbf{R}$) and constants. For example, when $n = 3, r = 3$ and $r' = 5$, the 5-th equation in pixel-level calibration is:

$$k_{11}h_{11} + k_{12}h_{12} + k_{13}h_{13} + \cdots + k_{33}h_{33} =$$
$$(R_{7,1}k_{11} + R_{7,2}k_{12} + R_{7,3}k_{13} + \cdots + R_{7,9}k_{33})h_{11}+$$
$$(R_{8,1}k_{11} + R_{8,2}k_{12} + R_{8,3}k_{13} + \cdots + R_{8,9}k_{33})h_{12}+$$
$$\cdots +$$
$$(R_{19,1}k_{11} + R_{19,2}k_{12} + R_{19,3}k_{13} + \cdots + R_{19,9}k_{33})h_{33} \tag{7}$$

We rearrange the equation and get:

$$(1 - R_{7,1})k_{11}h_{11} + (-R_{7,2})k_{12}h_{11} + \cdots (1 - R_{19,9})k_{33}h_{33} = 0. \tag{8}$$

To ensure this equation for all $k_{ij}h_{ij}$, one can let every coefficient be zero. Getting a linear equation set for the 5-th equation in pixel-level calibration.

$$\begin{cases} R_{7,1} = -1 \\ R_{7,2} = 0 \\ \qquad \cdots \\ R_{19,9} = -1 \end{cases} \tag{4}$$

We do this for every equation in Eq. (6) to derive a larger set of linear equations of $R_{ij}$. Note that the equation sets are linear equations of $R_{ij}$ and have no $h$ or $k$, making it applicable to all conv kernels and any input feature. Let's go back to the general case where $\mathbf{R} \in \mathbb{R}^{r'^2 \times r^2}$. Let $A_{\text{structure}}$ denote the coefficient of the linear combination for $R_{ij}$ in structure-level calibration, and let $b_{\text{structure}}$ denote the right-hand-side constants in the equation set of structure-level calibration. Similarly, we define $A_{\text{pixel}}, b_{\text{pixel}}$, and we construct a least square problem.

$$A\mathbf{x} = b, \quad A = \begin{bmatrix} A_{\text{structure}} \\ \eta A_{\text{pixel}} \end{bmatrix}, b = \begin{bmatrix} b_{\text{structure}} \\ \eta b_{\text{pixel}} \end{bmatrix}, \mathbf{x} = \begin{bmatrix} R_{1,1} \\ R_{1,2} \\ \cdots \\ R_{r'^2,r^2} \end{bmatrix}. \tag{5}$$

The solution is $\mathbf{x} = (A^T A)^{-1} A^T b$. This can be easily solved by math software, i.e., MATLAB.

## G  SEARCHING RE-DILATION SCHEDULE

We first test a series of hand-crafted hyperparameters and figure out some empirical results: 1) Using re-dilation/dispersion in inner U-Net blocks and using original convolution in outer U-Net blocks produces good results. 2) Sharing the same re-dilation scale in a block instead of using different re-dilation scales for every convolution layer within a block produces better results.

Then, we search for hyperparameters using an automatic strategy. For each target resolution, we sample 50 examples from LAION-5B to build an evaluation set. The MSE loss of noise estimation, used in the training of diffusion models, serves as a metric for the hyperparameter search at each diffusion timestep. The hyperparameter that achieves the lowest loss is chosen for the corresponding timestep.

To speed up the search process, we prune the hyperparameter set using the previously established empirical rules. The pruning method is as follows: 1) Use the same re-dilation scale in all convolution layers within a block. 2) The blocks within a start block and an end block will use re-dilation/dispersion. For example, SD v2.1 blocks are shown in Fig. 9. If the start block is DB2 and the end block is UB1, then the blocks that use re-dilation/dispersion are DB2, DB3, MB, UB0, and UB1. 3) If the search includes noise-damped classifier-free guidance, then the blocks that use re-dilation/dispersion in $\epsilon_\theta$ is a continuous subset of the blocks that use re-dilation/dispersion in $\tilde{\epsilon}_\theta$. For example, if $\tilde{\epsilon}_\theta$ re-dilation/dispersion blocks are DB2, DB3, MB, UB0, UB1, that in $\epsilon_\theta$ can be DB3, MB, UB0. 4) The maximum re-dilation scale does not exceed the enlargement scale of the target resolution (i.e., generating a 2048×2048 image using a 512×512 model, the maximum re-dilation scale is 4). The search step of the re-dilation scale is 1.

# H    OTHER VISUALIZATIONS

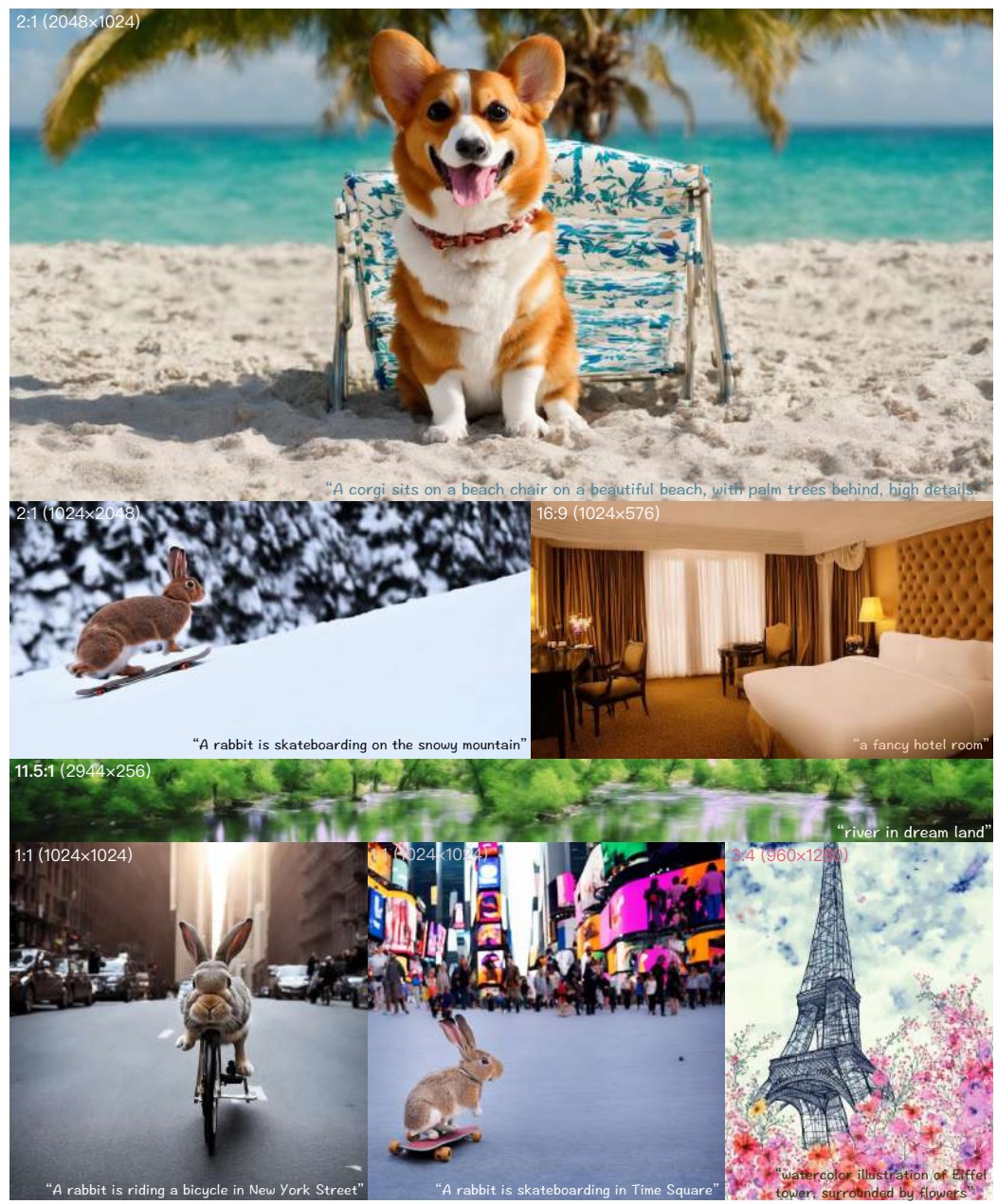

Figure 14: More generated results with our method and SD 2.1 with arbitrary aspect ratios and sizes.

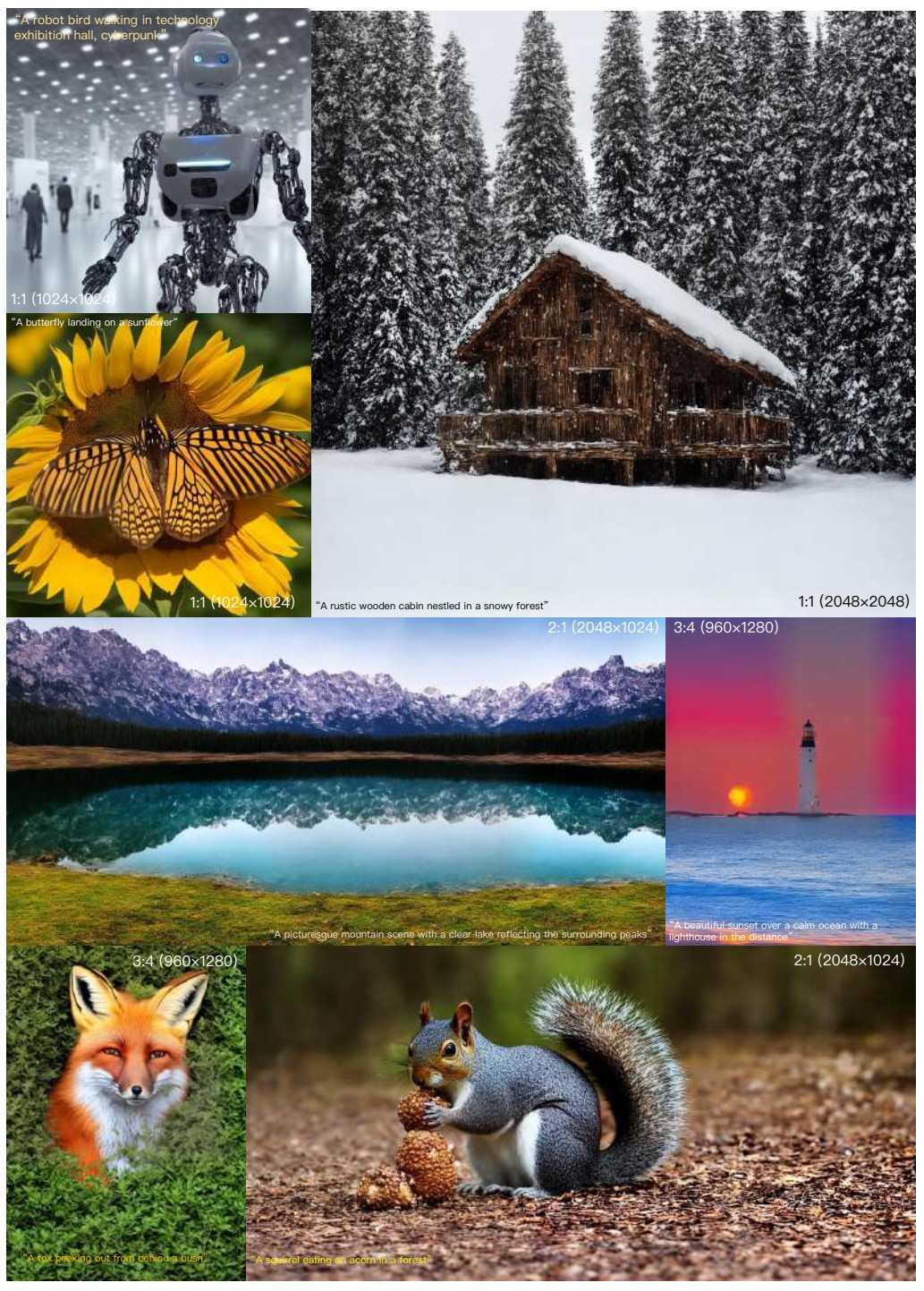

Figure 15: More generated results with our method and SD 1.5 with arbitrary aspect ratios and sizes.

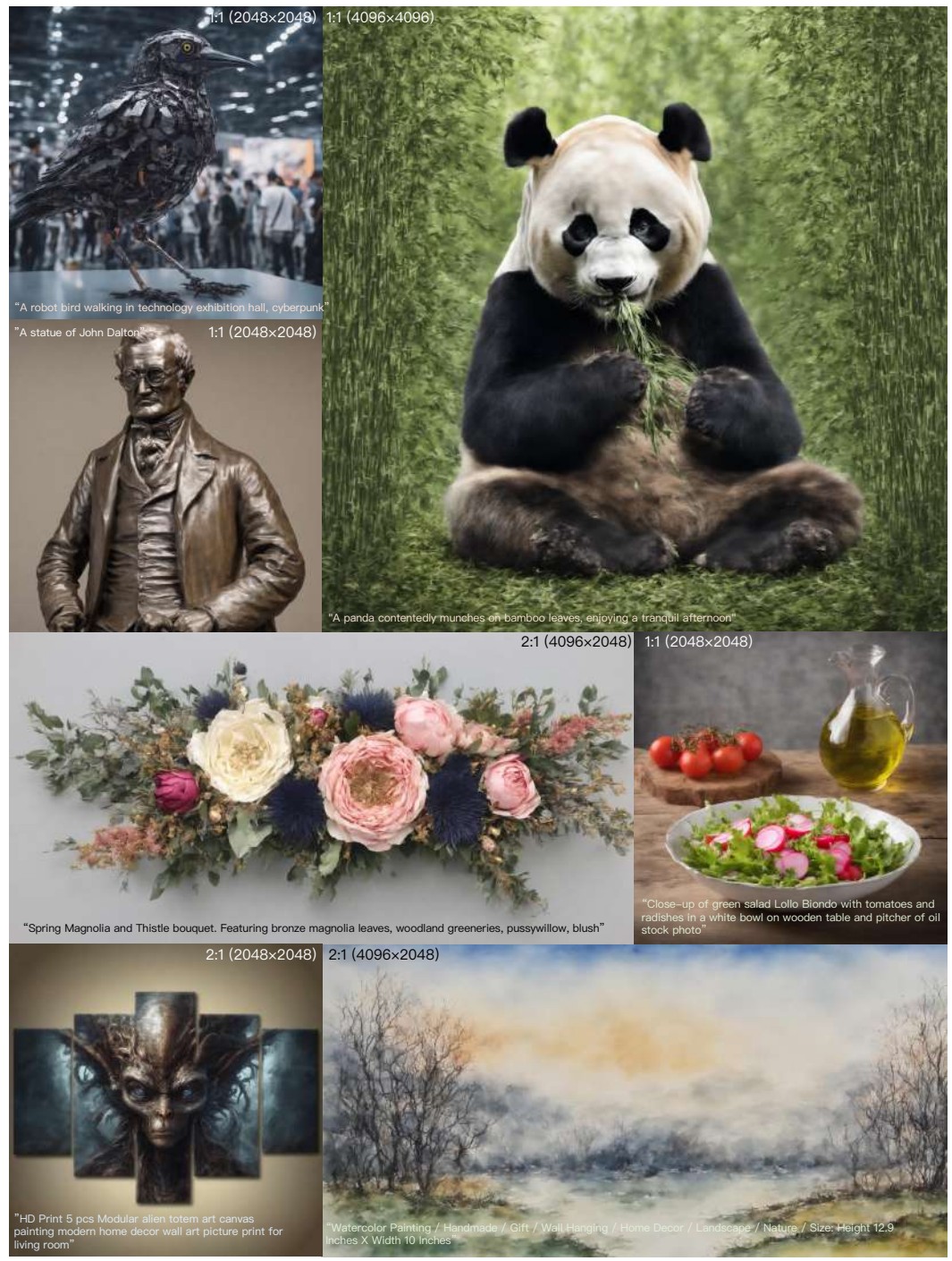

Figure 16: More generated results with our method and SD XL 1.0 with arbitrary aspect ratios and sizes.

