# OpenReview forum: "ScaleCrafter: Tuning-free Higher-Resolution Visual Generation with Diffusion Models"
_ICLR.cc/2024/Conference — ICLR 2024 spotlight_

### Official Review · Reviewer_3ixD · 2023-10-25

**Soundness:** 3 good
**Presentation:** 3 good
**Contribution:** 3 good
**Rating:** 6
**Confidence:** 4

**Summary:**

This paper proposes an (almost) training-free solution for adapting a pre-trained diffusion model to generate images of much higher resolution than the image size used during training.
To achieve this goal, the authors consider the procedure of convolution dispersion which aims to increase the receptive field (in terms of pixels) of convolutional blocks while preserving the properties of the original layer.
In addition, the presented technique of noise-damped classifier-free guidance leverages both the original model and its modification with dispersed convolutions in order to combine the generative power of the former and better denoising in high resolution of the latter.

According to the qualitative evaluation, the proposed approach successfully eliminates the issue of object repetition and implausible object structures. This is also confirmed with quantitative measurements.

**Strengths:**

The problem tackled in this work is quite important for the community. In general, this paper is clearly written. The motivation is explained well, and the details of the method are concise (see minor remarks below). The samples provided by the authors look quite plausible. Also, I appreciate the implementation details specified in the Appendix.

**Weaknesses:**

1. As far as I can judge, there is no discussion in the paper of the ability to magnify the output produced with the original diffusion model in standard resolution. I suggest adding it to the paper since it seems that it can be relatively straightforward to add it to the existing guidance (Eq. 5). This can help to make the comparison with SD+SR (Tab. 2) more conclusive.
1.  None of the reported quantitative metrics except for human assessment estimates the plausibility of generated high-frequency details, since common implementations of FID/KID downsample the input images to the ImageNet resolution before the feature extraction. Therefore, I recommend reporting e.g. patch-FID [1]  to assess high-resolution textures in addition. Also, some other variations like sFID [2] are known for better assessment of spatial variability.

1. Minor remarks:
    1.  Eq. 1 probably has a typo: the variable $o$ exists in LHR but not in RHS.
    1.  Probably, one needs to replace $\min$ with  $\arg\min$ in Eq. 4.
    1.  Please highlight the best metrics in tables consistently (see Tab. 2).
    1.  There is no such block as MB3 in Fig. 1 of Appendix A2, please edit the caption.

References:

[1] Chai et al. Any-resolution Training for High-resolution Image Synthesis. In ECCV, 2022.

[2] Nash et al. Generating images with sparse representations. In ICML, 2021.

**Questions:**

1. Please, address the weaknesses mentioned above.
1. Since the dispersed convolution is a dense operation with a larger receptive field, it introduces additional computational costs. How significant is the computational overhead of the method?
1. Is the re-dilation described in Sec. 3.2 actually used in the method? As far as I understood, in practice only dispersed convolutions are applied in the modified UNet. Probably, this part of the description needs more polishing.
1. Is it possible to find a closed-form solution for the optimization problem in Eq. 4? If so, it would be nice to add it to the paper or supplementary. Otherwise, the method implementation is not entirely training-free (although still pretty cheap), and the claims should be adjusted.

---

> ### Author Response · Authors · 2023-11-22
> **Response to Reviewer 3ixD [2/2]**
>
> > Q2. Since the dispersed convolution is a dense operation with a larger receptive field, it introduces additional computational costs. How significant is the computational overhead of the method?
>
> Please refer to our response to reviewer URVR Q2. Thank you!
>
> > Q3. Is the re-dilation described in Sec. 3.2 actually used in the method?
>
> Yes, the re-dilation is used in all the settings. However, when the resolution exceeds 1280x1280, we need to further incorporate the noise-damped classifier-free guidance (ndcfg), and the dispersed convolution. We have documented all these details in the hyperparameters section of the supplementary. We also clarify this in the revision of the main paper.
>
> > Q4. Is it possible to find a closed-form solution for the optimization problem in Eq. 4?
>
> **There is a closed-form solution for dispersed convolution. One only needs to solve a least square problem and no training is needed.** The detailed method is as follows.
>
> We follow the notation in our main paper. Given a convolution layer with kernel $\boldsymbol{k} \in \mathbb{R}^{r \times r}$ and a target kernel size $r'$. We find a dispersion transform $\mathbf{R} \in \mathbb{R}^{r'^2 \times r^2}$ to get a dispersed kernel $\boldsymbol{k}'$. Consider an input feature map $\boldsymbol{h} \in \mathbb{R}^{n \times n}$, we ignore the bias in convolution, since it will not influence our results.
>
> Structure-level calibration and pixel-level calibration can form an equation set
>
> $$\left\\{\begin{matrix}  interp_d (f_k(\boldsymbol{h})) = f_{\boldsymbol{k}'}(interp_d(\boldsymbol{h})) \\\ \eta f_{\boldsymbol{k}} (\boldsymbol{h}) = \eta f_{\boldsymbol{k}'}(\boldsymbol{h}). \end{matrix}\right.
>  \tag{1}$$
>
> $ interp_d (f_k(\boldsymbol{h})) \in \mathbb{R}^{nd \times nd}, f_{\boldsymbol{k}} (\boldsymbol{h}) \in \mathbb{R}^{n \times n} $.  The equation set has $(nd)^2 + n^2$ equations. Each equation is made up of the sum of terms $k_{ij}h_{ij}$ (elements in $\boldsymbol{k}$ and $\boldsymbol{h}$) where the coefficient of the terms are linear combinations of $R_{ij}$ (elements in $\mathbf{R}$) and constants. For example, when $n=3, r=3$ and $r'=5$, the 5-$th$ equation in pixel-level calibration is
> $$
> \begin{matrix} k_{11} h_{11} + k_{12} h_{12} + k_{13} h_{13} + \cdots + k_{33} h_{33} = \\\ (R_{7,1} k_{11} + R_{7,2} k_{12} + R_{7,3} k_{13} + \cdots + R_{7,9} k_{33})h_{11} +  \\\  (R_{8,1} k_{11} + R_{8,2} k_{12} + R_{8,3} k_{13} + \cdots + R_{8,9} k_{33})h_{12} + \\\ \cdots + \\\ (R_{19,1} k_{11} + R_{19,2} k_{12} + R_{19,3} k_{13} + \cdots + R_{19,9} k_{33})h_{33} \end{matrix}
> \tag{2}$$
> We rearrange the equation and get
> $$\begin{align}
> (1 - R_{7,1}) k_{11} h_{11} + (-R_{7,2})k_{12} h_{11} + \cdots (1 - R_{19, 9})k_{33}h_{33} = 0 \notag.
> \end{align} \tag{3} $$
> To ensure this equation for all $k_{ij}h_{ij}$, one can let every coefficient be zero. Getting a linear equation set for the 5-$th$ equation in pixel-level calibration:
> $$ \left\\{\begin{matrix} R_{7, 1} = -1 \\\ R_{7, 2} = 0 \\\ \cdots \\\ R_{19, 9} = -1
> \end{matrix} \right. \tag{4}$$
> We do this for every equation in Eqn.(1) to derive a larger set of linear equations of $R_{ij}$. Note that the equation sets are linear equations of $R_{ij}$ and have no $\boldsymbol{h}$ or $\boldsymbol{k}$, making it applicable to all conv kernels and any input feature. Lets back to general case where $\mathbf{R} \in \mathbb{R}^{r'^2 \times r^2}$. Let $A_{\mathrm{structure}}$ denote the coefficient of linear combination for $R_{ij}$ in structure-level calibration and let $b_{\mathrm{structure}}$ denote the right-hand-side constants in the equation set of structure-level calibration. Similarly, we define $A_{\mathrm{pixel}}, b_{\mathrm{pixel}}$. We construct a least square problem
>
> $$A\mathbf{x} = b, \quad A = \left[\begin{matrix}A_{\mathrm{structure}} \\\ \eta A_{\mathrm{pixel}}\end{matrix} \right], b = \left[\begin{matrix}b_{\mathrm{structure}}  \\\ \eta b_{\mathrm{pixel}}\end{matrix} \right], \mathbf{x} = \left[\begin{matrix} R_{1, 1} \\\ R_{1, 2} \\\ \cdots \\\ R_{r'^2, r^2}\end{matrix}\right]. \tag{5}$$
>
> The solution is $\mathbf{x} = (A^T A)^{-1} A^T b$. This can be easily solved by math software, i.e., MATLAB. We have included this part in the supplementary materials.

---

> > ### Comment · Reviewer_3ixD · 2023-12-01
> >
> > I would like to thank the authors for their work during the rebuttal period and for the provided clarification. I keep my initial score.

---

### Official Review · Reviewer_URVR · 2023-10-30

**Soundness:** 3 good
**Presentation:** 3 good
**Contribution:** 3 good
**Rating:** 6
**Confidence:** 3

**Summary:**

This paper conduct research on the capability of generating higher resolution image from pre-trained diffusion models using lower resolution training data. The authors find that in the existing structures, the perception field of convolutional kernels is limited. Based on this, the authors propose a re-dilation method that can adjust the convolutional perception field during inference. The proposed method in this paper can generate high resolution images with 4096x4096 resolution without extra training. Experimental results show that the proposed method is able to achieve the SOTA results. This paper is well organized and easy to follow.

**Strengths:**

Here are the strength points of this paper:

The authors proposed an observation and explanation for the objective repetition problem. They also designed a re-dilation method to address this problem for high-resolution image generation. The proposed method is applied and evaluated on multiple models, the results show that the proposed method is effective.

**Weaknesses:**

Here are the weak points of this paper:

See the detailed comments and questions.

**Questions:**

Here are my detailed comments and suggestions:

This article is generally good, but I still have some questions about the design of the model and the experimental process.

1.	The section on "re-dilation" mentions the use of a predefined dilation schedule function D(t,l). How is this function defined, and have any experiments been conducted to validate or optimize its selection?

2.	In the "convolution dispersion" section, a linear transformation R and structural-level calibration are used to enlarge the convolution kernel. Does this step add to the model's computational complexity? If so, by how much?

3.	In the "noise-damped classifier-free guidance" section, a guidance scale w is mentioned. How is this parameter chosen? Does it require adjustments for different tasks or datasets?

4.	The authors used FID and KID as the metrics that require downsampling images to 229 × 229, does this imply that these metrics are not very suitable for evaluating high-resolution images?

5.	Has the paper considered comparisons with other recently published methods for high-resolution image generation? Would such comparisons further validate the effectiveness of the model?

---

> ### Author Response · Authors · 2023-11-22
> **Response to Reviewer URVR**
>
> Thank you for providing useful feedback. Below are our responses for each point.
>
> > Q1. The section on "re-dilation" mentions the use of a predefined dilation schedule function D(t,l). How is this function defined, and have any experiments been conducted to validate or optimize its selection?
>
> We define this function via both manual effort and automatic hyperparameter search.
> Once the hyperparameters are settled on a very few images, they can be used for all other samples.
>
> We introduce the details of the re-dilation schedule as follows:
> We first test a series of hand-crafted hyperparameters and figure out some empirical rules:
> + Using re-dilation/dispersion in inner UNet blocks and using original convolution in outer UNet blocks produces good results. Because outer blocks are responsible for the low-level denoising rather than keeping the geometry structures.
> + Sharing the same re-dilation scale in a block instead of using different re-dilation scales for every convolution layer in a block produces better results.
>
> Then, we search the hyperparameters using an automatic strategy. For each target resolution, we sample 50 examples from LAION-5B to build up an evaluation set. The MSE loss of noise estimation used in the training diffusion models is used to serve as a metric for the hyperparameter search at each diffusion timestep.
> The hyperparameter that achieves the least loss is chosen for the corresponding timestep.
>
> We have included this part in detail in the supplementary materials.
>
> > Q2. In the "convolution dispersion" section, a linear transformation R and structural-level calibration are used to enlarge the convolution kernel. Does this step add to the model's computational complexity? If so, by how much?
>
> Dispersed convolution leads to slight computation overhead which is due to the quadratic complexity $O(N^2)$ of the convolution operator where $N$ is the kernel size. Therefore, we suggest using dispersed kernels as small as possible. In our case, we enlarge $3\times 3$ convolution kernels to $5 \times 5$. Theoretically, it leads to 2.7x more convolution computation in UNet. However, in practice, not all convolution layers need dispersion, and the parallel computing of GPUs will alleviate that overhead. We compare the overhead with and without $5\times 5$ dispersed convolution on $2048\times 2048$ text-to-image with SD v1.5 on one A100 40G GPU.
>
> ||Time per iteration|Memory usage|
> |-|-|-|
> |w/o dispersion|2.47s|6,482MB|
> |w/ dispersion|2.78s|9,874MB|
> |Overhead|1.13x|1.52x|
>
> > Q3. In the "noise-damped classifier-free guidance" section, a guidance scale w is mentioned. How is this parameter chosen? Does it require adjustments for different tasks or datasets?
>
> The choice of the scale parameter, w, follows the same scale used in the classifier-free guidance (cfg) of the original model. Therefore, there is no need for us to adjust the value of w ourselves. Different tasks and datasets may require different values of w for the cfg. The w in our noise-damped classifier-free guidance (ndcfg) can just remain the same as in the standard cfg.
>
> >Q4. The authors used FID and KID as the metrics that require downsampling images to 229 × 229, does this imply that these metrics are not very suitable for evaluating high-resolution images?
>
> FID and KID can only be used to measure the overall generation quality such as the global structure. Due to the downsampling issue, they are not able to reflect the texture details and definition.
> Thus we add four extra metrics pFID, pKID, sFID, and sKID to address this problem. Please refer to the general response for the results, which show that our method achieves significantly better sFID, and sKID scores and highlight the superiority of our method on the texture details.
>
> >Q5. Has the paper considered comparisons with other recently published methods for high-resolution image generation? Would such comparisons further validate the effectiveness of the model?
>
> Thank you for your suggestion. We compare our method with SD XL for high-resolution image generation. Please refer to the general response. The results show that our method via SD 2.1 surpasses the SD XL on the subset of Laion 5B and has a faster inference speed and fewer model parameters.

---

> > ### Comment · Reviewer_URVR · 2023-12-04
> >
> > Thanks for the author's response. The authors have addressed all my concerns.

---

### Official Review · Reviewer_6a3j · 2023-10-31

**Soundness:** 4 excellent
**Presentation:** 3 good
**Contribution:** 4 excellent
**Rating:** 8
**Confidence:** 4

**Summary:**

This submission proposes to generate images from pre-trained diffusion models at a much higher resolution without the problems of object repetition and unreasonable object structures. In particular, it presents the key problem of the limited perception field of convolutional kernels and represents the new dispersed convolution and noise-damped classifier-free guidance. The experimental results, including ultra-high-resolution image and video synthesis, are very impressive.

**Strengths:**

- The paper addresses a significant challenge in image generation by investigating the generation of images at much higher resolutions than the training image sizes, while also allowing for arbitrary aspect ratios. This is a novel and important problem in the field of image synthesis.
- The proposed approach of using re-dilation to dynamically adjust the convolutional perception field during inference is a key contribution. This addresses the identified limitation of convolutional kernels, which is a crucial insight for improving high-resolution image generation.
- The paper's extensive experiments demonstrate the effectiveness of the proposed approach in addressing the repetition issue and achieving state-of-the-art performance, particularly in texture details. This provides strong empirical support for the proposed techniques.
- Overall, the paper appears to make a valuable contribution to the field of image synthesis, specifically in the context of generating high-resolution images with arbitrary aspect ratios. The proposed techniques and insights are well-motivated and supported by extensive experiments.

**Weaknesses:**

- Fig. 5-7 and Table 2-4 could be presented in a former manner.
- More ablation studies could be presented in the main text and limitations are suggested.

**Questions:**

Please refer to the above weaknesses.

**Details Of Ethics Concerns:**

N.A.

---

> ### Author Response · Authors · 2023-11-22
> **Response to Reviewer 6a3j**
>
> Thank you for your time to review our paper and the encouraging comments!
>
> > W1. Fig. 5-7 and Table 2-4 could be presented in a former manner.
>
> Thank you for your suggestion! We have made revisions to the illustration and also reorganized the order of these components accordingly.
>
> > W2. More ablation studies could be presented in the main text, and limitations are suggested.
>
> Thanks for your suggestion! We have replenished more ablations and limitations in the supplementary due to the page limit of the main text. Please refer to Sections D and E in the supplementary materials for detailed information on the ablations and limitations.

---

### Official Review · Reviewer_7LTr · 2023-11-03

**Soundness:** 2 fair
**Presentation:** 3 good
**Contribution:** 2 fair
**Rating:** 6
**Confidence:** 4

**Summary:**

This submission proposed the generation of high-resolution images from pre-trained diffusion models. The authors claimed that their approach can address the persistent issues with the generated images. A re-dilation method was proposed to dynamically adjust the convolutional perception field during inference. Moreover, the authors implemented ultra-high-resolution image generation with a dispersed convolution and noise-damped classifier-free guidance. The experiments were conducted to demonstrate the advantages of the proposed solutions.

**Strengths:**

As claimed by the authors, the proposed solution is simple and efficient, but it is powerful enough to generate ultra-high-resolution images and videos. The method does not require any training and optimization.

**Weaknesses:**

The major issue is with the experiments. It is unsure whether the results in Table 1 tell about the differences among SD solutions or the differences between the "Method" in the second column. And when the resolution changes, how did the results vary accordingly? Are the metrics still acceptable if a higher-resolution image is generated? What do the metrics mean for the application in addition to their use for comparison?

As the authors mentioned the results in Table 2 were not better than the SR+SR method, what is the justification to use the proposed solution? How could the user balance the efficiency and the quality in terms of the application's needs? It is unclear.

**Questions:**

The significance of the proposed method is still unclear after the presentation of the experimental results. The method can achieve similar or slightly better results. However, it is not sufficient to justify the significance of the work. This is a major concern.

---

> ### Author Response · Authors · 2023-11-22
> **Response to reviewer 7LTr**
>
> Thank you for providing valuable feedback. We have carefully considered your questions and made corresponding revisions and experiments.
>
> > Q1. The significance of the proposed method is still unclear after the presentation of the experimental results.
>
> Let us first clarify the significance of our work:
>
> Our main focus is to uncover the reason behind the structural distortion observed when directly inferencing SD at higher resolutions, which is attributed to the convolutional receptive field. **The key significance is the observation of the problem and the findings of its essential cause.**
>
> Based on these crucial observations, we surprisingly found altering the network receptive field, only during inference, can effectively generate higher-resolution images and videos. We believe our insights may **inspire future research in designing more effective and efficient super-resolution or high-resolution image and video models**.
>
> From the application view, our approach does not require any training and big datasets, and **can be directly applied to the video generation as a free solution**. Furthermore, following your suggestions, we provide a comprehensive study on the general generation quality, texture details, inference time, and model parameters in Table 1 of the general response (also documented in Supplementary Table 10), showing **better texture details, better efficiency, and better generation performance (when using low-resolution as guidance during the inference) of our method compared with SD+SR**.
>
> It is worth mentioning that the significance of our findings has also been acknowledged by other reviewers, who recognized that our work **addresses a significant challenge and tackles a novel and important problem** in image synthesis (reviewer 6a3j). It was also highlighted by reviewer 3ixD, who considered it **quite important to the research community**.
>
> > W1. The major issue is with the experiments. It is unsure whether the results in Table 1 tell about the differences among SD solutions or the differences between the "Method" in the second column. And when the resolution changes, how did the results vary accordingly? Are the metrics still acceptable if a higher-resolution image is generated? What do the metrics mean for the application in addition to their use for comparison?
>
> 1. Purpose of Table 1: Table 1 aims to compare the performance of our method with the baseline methods for the problem of sampling higher-resolution images than the training resolution, rather than comparing different SD solutions. We have made the necessary modifications in the caption of Table 1 to convey our intent.
> 2. Impact of resolution changes: The metrics deteriorate as the desired resolution increases as the setting becomes more and more challenging. However, our method consistently outperforms all other baselines. For the visual results, please refer to Section D in supplementary materials for the results with increasing resolutions.
> 3. Acceptability of metrics: The metrics are acceptable and can be checked via the visual results provided on our anonymous website. Note that due to the long inference time of high-resolution images, we calculate the metrics on 10k images on the settings of 6.25x, 8x, and 16x, and 30k images on the settings of 4x. Thus, we do not intend to compare the performance across different resolutions.
> 4. Meaning for the application: The lower scores of FID_real and KID_real mean better general generation quality and FID_base and KID_base mean the performance gap when inferencing higher-resolution images than the training resolution, indicating the scale-generalization ability of the model. Additionally, we have also incorporated two additional metrics (patch-FID/KID and sFID/KID) to evaluate the quality and sharpness of texture details in Table 1 of the general response to better evaluate our method from the application aspect.
>
> > W2. As the authors mentioned the results in Table 2 were not better than the SR+SR method, what is the justification to use the proposed solution? How could the user balance the efficiency and the quality in terms of the application's needs?
>
> Justification for using our proposed solution compared with SD+SR:
>
>  1. Please check Table 1 in the general response where we compare our method with SD+SR comprehensively including both efficiency and quality.
>
>  2. Moreover, there is currently no open-domain video SR model available for high-resolution video generation. Applying the image SR model on video leads to the temporal jitter issue because the details generated for each frame are independent. In contrast, our method addresses this issue by generating higher-resolution videos while leveraging the temporal modeling ability inherited from the lower-resolution pretrained model. This allows us to generate higher-resolution videos via a free solution.
>
> We sincerely appreciate your valuable feedback and hope that our response addresses your concerns.

---

### Author Response · Authors · 2023-11-22
**Official general response**

We thank all reviewers for their valuable feedback.

Before addressing the concerns mentioned in the reviews, we would like to emphasize the advantages of our work that have been recognized by the reviewers:
Reviewer 6a3j acknowledged that our work addresses **a significant challenge and tackles a novel and important problem** and highlighted the **crucial insights and valuable contributions** our work brings to the field.
Reviewer 3ixD also acknowledged that **the problem tackled in this work is quite important for the community**.
Furthermore, our method has been praised for its **simplicity, efficiency** (reviewer 7LTr), and **effectiveness** (reviewer URVR).
The **impressive results** we have achieved have also been acknowledged by reviewer 6a3j.
We greatly appreciate the reviewers's affirmations and positive feedback on our work!

**Here, we address several concerns comprehensively**, including the advantages of our method compared with the pretrained super-resolution (SR) method (reviewer 7LTr), the comparison with the high-resolution method (reviewer URVR), the incorporation of low-resolution guidance (LR)  to make a fair comparison with SR (reviewer 3ixD), metrics for measuring details (reviewer URVR, 3ixD), as well as the efficiency and quality aspects (reviewer 7LTr).

| Method         | FID$_r$ | KID$_r$ | pFID$_r$ | pKID$_r$ | sFID$_r$ | sKID$_r$ | Time (s) | #param |
|----------------|---------|---------|----------|----------|----------|----------|----------|-----------|
| SD XL          | 18.50   | 0.005   | 29.63    | 0.014    | 16.68    | 0.007    | 6.5      | 3.5B      |
| SR with SD 2.1     | 15.39   | 0.005   | **17.30** | **0.005** | 14.57    | 0.007    | 9.5 (1.5+8) | 2.2B      |
| Ours (SD 2.1)  | 18.73   | 0.005   | 20.97    | **0.005** | **10.17** | **0.004** | **5.6**  | **1.3B**  |
| Ours (SD 2.1)+ LR | **9.96** | **0.003** | 19.27    | 0.007    | 11.05    | **0.004** | 6.3 (1.5+4.8) | **1.3B**  |

Table 1: Compared with the state-of-the-art image generation model SD XL and super-resolution (SR) model under the resolution of 1024$^2$. `pFID_r` and `pKID_r` represent patch-FID and patch-KID, respectively, computed on real images. FID and KID are used to measure the overall generation performance, and pFID[1], pKID[1], sFID[2], and sKID[2] are used to measure the texture details of generated samples. Specifically, pFID/pKID avoids the downsampling operation and instead uses cropping in the metric calculation. sFID/sKID uses the features before the global average pooling to retain low-level details in the feature for the metric calculation, as well as avoids downsampling. `Time` indicates the seconds used for synthesizing one image on one A100 40G GPU with 16-bit precision. `#param` stands for the number of model parameters. The evaluation dataset is a 30k subset from Laion-5B with a resolution larger than 1024$^2$.

Results show that our training-free method (with no low-resolution reference image) achieves almost the comparable generation performance compared with well-trained SD+SR. Additionally, we achieve better texture details than SD + SR (see the sFID and sKID metrics, as well as the user study in the main paper). With the low-resolution generated samples as guidance, our method achieves much better results than SD+SR. At the same time, our method has 59% inference time and 59% model parameters less than SD+SR, showing our better efficiency. Compared with SD XL, we achieve both better metrics and lower inference time and parameter numbers. We replenish this part in Section C of Supplementary.

[1] Chai et al. Any-resolution Training for High-resolution Image Synthesis. In ECCV, 2022.

[2] Nash et al. Generating images with sparse representations. In ICML, 2021.

---

### Meta-Review · Area_Chair_wksv · 2023-12-05

**Metareview:**

The authors attempt to generate images at a substantially higher resolution than was seen during training and find that direct inference techniques lead to object repetition and unreasonable object structures. Other techniques that address doing joint diffusion do not address the problem since, as the authors find, they lack the correct receptive fields in order to fully analyze the full image's structure. To fix this, the authors propose a re-dilation of the convolutions during inference time (either at integer values or fractional values) to handle 2-4x upsamples and "dispersing" the convolution to handle even larger upsamples, 8x-16x. In addition, the authors find that at 16x, applying the dispersion to the outside convolutions causes a degradation in the noise estimation where they mix a re-dilated or dispersed block with a normal model. The authors show quantitative and qualitative evidence that their technique works over baselines.

**Justification For Why Not Higher Score:**

Between spotlight and oral, not completely sure. I think there could have been more analysis on the limits of the technique (presumably at some point it breaks) and the comparison with the image SR model.

**Justification For Why Not Lower Score:**

The results are impressive on an important problem (higher resolution generation). The paper has a clear insight and follows it directly with a technique that demonstrates the power of the insight.

---

### Decision · Program_Chairs · 2024-01-16

Accept (spotlight)